# Maternal Pre-Pregnancy BMI and Gestational Weight Gain Modified the Association between Prenatal Depressive Symptoms and Toddler’s Emotional and Behavioral Problems: A Prospective Cohort Study

**DOI:** 10.3390/nu15010181

**Published:** 2022-12-30

**Authors:** Shumin Zhang, Xuemei Ma, Qian Wei, Yunhui Zhang, Ling Wang, Huijing Shi

**Affiliations:** 1Department of Maternal, Child and Adolescent Health, School of Public Health, Fudan University, No. 130, Dong’an Road, Shanghai 200032, China; 2Department of Environmental Health, School of Public Health, Fudan University, No. 130, Dong’an Road, Shanghai 200032, China

**Keywords:** maternal prenatal depressive symptoms, body mass index, gestational weight gain, behavior problems, toddlers

## Abstract

Background: Maternal prenatal depressive symptoms and abnormal pre-pregnancy BMI have been scarcely reported to play interactive effects on child health. In this prospective cohort, we aimed to examine the interactive effects of maternal prenatal depressive symptoms and pre-pregnancy BMI as well as gestational weight gain (GWG) on offspring emotional and behavioral problems (EPBs). Methods: The study samples comprised 1216 mother–child pairs from Shanghai Maternal–Child Pairs Cohort recruited from 2016 to 2018. Maternal pre-pregnancy BMI and GWG were obtained from medical records, and maternal depressive symptoms were assessed via the Center for Epidemiological Studies Depression Scale (CES-D) at 32–36 gestational weeks. The child completed the behavioral measurement via the Strengths and Difficulties Questionnaire (SDQ) at 24 months postpartum. Results: There were 12.01% and 38.65% women with prenatal depressive symptoms and sub-threshold depressive symptoms during late pregnancy. Both maternal depressive symptoms and prenatal sub-threshold depressive symptoms were associated with higher internalizing (OR = 1.69, 95% CI, 1.05–2.72; OR = 1.48, 95% CI, 1.06–2.07) and externalizing (OR = 2.06, 95% CI, 1.30–3.25; OR = 1.42, 95% CI, 1.02–1.99) problems in children. Maternal pre-pregnancy BMI and GWG modified the association between prenatal depressive symptoms and child externalizing or total difficulties problems (*p* < 0.10 for interaction). Among the overweight/obese pregnant women, maternal prenatal depressive symptoms were associated with a higher risk of externalizing problems (OR = 2.75, 95% CI, 1.06–7.11) in children. Among the women who gained inadequate GWG, maternal prenatal sub-threshold depressive symptoms were associated with 2.85-fold (95% CI 1.48–5.48) risks for child externalizing problems, and maternal depressive symptoms were associated with higher externalizing and total difficulties problems (OR = 4.87, 95% CI, 2.03–11.70 and OR = 2.94, 95% CI, 1.28–6.74, respectively), but these associations were not significant in the appropriate or excessive GWG group. Conclusions: Both maternal prenatal sub-threshold depressive symptoms and depressive symptoms increased the risks of child internalizing and externalizing problems at 24 months of age, while the effects on child externalizing problems were stronger among overweight/obese or inadequate GWG pregnant women. Our study highlights the importance of simultaneously controlling the weight of pregnant women before and throughout pregnancy and prompting mental health in pregnant women, which might benefit their offspring’s EBPs.

## 1. Introduction

Emotional and behavioral problems (EBPs) are a major health problem faced by children, adolescents, and even adults worldwide, which are always classified by internalizing problems (e.g., emotional problems, peer problems) and externalizing problems (e.g., aggressive and hyperactive behavioral problems, inattention, opposition, and defiance) in children. There are considerable numbers of children with one or more behavioral problems worldwide, especially in low- and middle-income countries or areas [1]. In a large cohort of 71,929 individuals aged 6~16 years in China, 17.6% of the children and adolescents were identified as having EBPs [2]. During the COVID-19 pandemic, the prevalence in children’s EBPs even increased, with more than 30% of Chinese children showing symptoms of irritation and inattention [3]. Previous studies have demonstrated associations between children’s EBPs and a variety of health problems including overweight [4], sleep disturbance [5], decline in learning ability [6], and increase burden of sickness absence (SA), and disability pension (DP) in young adulthood [7]. The continuity of EBPs in childhood and/or adolescence into adulthood is well-acknowledged [8]. For instance, the developmental trajectories of externalizing behaviors mostly predicted intrusive, aggressive, and rule-breaking behavior in adulthood [9], and depressive symptoms, one of the internalizing behaviors, are strongly persistent and have the potential to differentiate into anxiety in young adulthood [10]. Given the immense societal and individual burden of EBPs, the identification of their risk factors is of paramount importance for developing primary preventive strategies.

Maternal prenatal mental health [11] has been identified as potential risk factors for children’s EBPs. Substantial evidence suggests that maternal prenatal depressive symptoms could increase the risk for EBPs in their offspring individually such as emotional symptoms, peer problems, psychosocial difficulties, autism or developmental delay diagnosis, and so on [12,13,14,15,16]. However, most of the research is still limited to explore the effectiveness of maternal major depression disorder on EBPs in offspring, and there are few studies reflecting the effectiveness of sub-threshold depressive symptoms. Sub-threshold depressive symptoms are defined as presenting with some depressive symptoms for 2 weeks or longer (e.g., depressed mood or anhedonia) but not yet meeting the diagnostic criteria for a major depression disorder [17]. Sub-threshold depressive symptoms have been shown to be an important predictor of major depression disorder, thus putting the well-being of individuals with sub-threshold depressive symptoms at risk [18]. For example, an existing study also found that the overall impact on the mortality of sub-threshold depressive symptoms and major depression disorder was comparable, with the population attributable fraction being 7.0% [19]. Although the adverse effects of the illness are greater for people with clinical depression than for people with sub-threshold depressive symptoms, sub-threshold depressive symptoms accounts for a larger proportion in the population. Moreover, maternal sub-threshold depressive symptoms during pregnancy may have adverse effects on the health of their offspring [20], but there has been a relative lack of consensus in the literature. Therefore, focusing on the impact of sub-threshold depressive symptoms during pregnancy on the health of the offspring could provide a scientific basis and evidence for early intervention to achieve greater benefits.

Maternal abnormal pre-pregnancy BMI and inappropriate GWG are other common factors in the context of child development [21,22]. The existing literature from animals and humans has reported adverse effects of maternal pre-pregnancy overweight/obesity on child EBPs. For maternal GWG, which affects the neurobehavior of offspring, remains controversial. For example, some studies have reported that excessive GWG in women with pre-pregnancy overweight/obesity increased the risk of attention deficit and hyperactivity disorder (ADHD) symptoms [23] and problem behaviors in offspring [24], while the study of Pugh et al. [15] did not support these findings.

Recently, Cattane et al. pointed out that maternal overweight/obesity and depressive symptoms might impact child health in an interactive way [25] due to the various sharing mechanisms including hypothalamic–pituitary–adrenal (HPA) axis disorders, inflammatory factors, and metabolic disorders [26,27]. For example, previous studies have indicated that obese women are more vulnerable to developing prenatal and/or postnatal depressive symptoms than normal weight women during pregnancy [28]. Therefore, these factors should not be considered in isolation in the context of offspring neurodevelopment. However, evidence from populations in this context is still scarce. In a large-sample (*n* = 70,605) prospective cohort study, maternal overweight/obesity and depressive symptoms increased the higher risk of adverse birth outcomes combined than them alone [29]. However, the interaction effects of these factors on the offspring’s psychiatric problems in early childhood remain unclear. Additionally, considering the link between maternal overweight/obesity and GWG, the combined effects of maternal GWG and depressive symptoms should also be elucidated.

Therefore, prospective studies investigating the interplay of maternal prenatal depressive symptoms and abnormal weight status on child EBPs are urgently needed. Based on a prospective birth cohort, we aimed to investigate the association between maternal prenatal depressive symptoms and toddler’s EBPs at 24 months old, and explore the modified effects of pre-pregnancy BMI and GWG on the associations above.

## 2. Methods

### 2.1. Subjects and Design

The participants included mother–child dyads from the ongoing Shanghai Maternal–Child Pairs Cohort (Shanghai MCPC) [30], a large-scale prospective birth cohort in China, which recruited pregnant women during early pregnancy from April 2016 to December 2018 in two regional hospitals of Shanghai’s Pudong and Songjiang Districts. The inclusion and exclusion criteria that were established for cohort recruitment have been elaborated in prior studies based on the Shanghai MCPC cohort study [31]. The other inclusion criteria in this study were as follows: toddlers who were singleton live birth, toddlers whose mothers were over 18 years of age at delivery, toddlers whose mothers who had complete medical record of pre-pregnancy weight, height, and weight before delivery, and toddlers whose parents completed SDQ assessments in the follow-up at 24 months postpartum before June 2020. A total of 2710 singleton children whose mothers had medical records of pre-pregnancy weight, height, weight before delivery and completed the Center for Epidemiological Studies Depression Scale (CES-D) assessments at 32~36 gestational weeks. Among these 2710 children, 1216 completed the SDQ assessment at 24 months of age. Finally, data of 1216 mother–child dyads were used in the present study. This research was approved by the Institutional Review Board in Public Health School of Fudan University (IRB number 2016-04-0587, 2016-04-0587-EX). Written informed consent was also provided by all participants and the completion of study was voluntary throughout the follow-up.

### 2.2. Studied Variables

#### 2.2.1. Prenatal Depressive Symptoms

Maternal prenatal depressive symptoms over the two weeks were assessed via the Center for Epidemiological Studies Depression Scale (CES-D) [32] at 32–36 gestational weeks. The CES-D consists of 20 items and has been widely used for screening depressive symptoms in pregnant women [33]. A higher score of CES-D means more severe depressive symptoms. This study focused on maternal depressive symptoms rather than the full clinical diagnosis of mental health, and due to the adverse effects of sub-threshold depressive symptoms (the score of CES-D ranges from 8 to 15 points) on health [19], the women in the present study were categorized as the non-depressive symptoms group (CES-D score <8), the sub-threshold depressive symptoms group (CES-D score ranging from 8 to 15 points), and the depressive symptoms group (CES-D score ≥16) [19,34].

#### 2.2.2. Maternal Pre-Pregnancy BMI and GWG

Maternal pre-pregnancy weight, height, and weight before delivery were collected from medical records. Maternal pre-pregnancy weight was self-reported by the pregnant women and the height and weight before delivery were measured by nurses. Pre-pregnancy BMI and GWG were categorized as follows. Maternal pre-pregnancy BMI was calculated by dividing weight by height squared and grouped into underweight (<18.5 kg/m^2^), normal weight (18.5~23.9 kg/m^2^), and overweight/obese (≥24 kg/m^2^), according to the BMI criteria recommended by the Working Group on Obesity in China [35]. GWG was grouped as inadequate GWG (below the ranges), appropriate GWG (within the ranges), and excessive GWG (above the ranges) according to the pre-pregnancy BMI referring to the IOM recommended ranges [36], and the ranges of appropriate GWG in the underweight group, normal weight group, overweight group, and obese group were 12.5–18.0 kg, 11.5–16.0 kg, 7.0–11.5 kg, and 5.0–9.0 kg, respectively [37].

#### 2.2.3. Child Emotional and Behavioral Problems

The child EBPs were assessed using the Strengths and Difficulties Questionnaire (SDQ) filled by the parents at 2 years of old [38]. The SDQ, one of the most commonly used tools for screening children’s sub-clinical psychological and mental disorders, has been adapted to different versions depending on the language and custom, and the Chinese version has been adapted and widely used in the Chinese population [39,40]. In the current study, we used the 20-item scale measures of child difficulties in four dimensions including emotional symptoms, conduct behavior, hyperactivity behavior, and peer problems. The total difficulties score consists of the summed scores of the four dimensions. The internalizing score consists of the summed scores of the emotional symptoms and peer problems, while the externalizing score consists of the summed scores of the conduct behavior and hyperactivity behavior [41]. Toddlers were classified into the EBP group if their scores were up to the top 20% of scores, otherwise, the toddlers were not in the EBP group [42].

#### 2.2.4. Covariates

Covariates were selected based on prior evidence and statistically significant variables between groups such as maternal sociodemographic characteristics (age, education, annual household income), alcohol use, passive cigarette smoking, gestational complications, parity, delivery mode, offspring’s sex, preterm, and breastfeeding practice at 6 months old. Maternal characteristics such as sociodemographic information (age, education, annual household income), pre-pregnancy alcohol use, passive cigarette smoking, and breastfeeding practice at 6 months old were collected by questionnaire. Data on gestational complications, parity, delivery mode, and preterm were obtained from medical records. Gestational complications were defined as one or more of gestational diabetes mellitus, gestational hypertension, preeclampsia, or threatened abortion among the pregnant women [43,44]. Maternal educational level and annual household income were standardized separately and added up to generate the total standardized score, namely social economic status (SES). Then, according to the mean segmentation, the total standardized scores of SES were recorded as a dichotomous variable (high SES and low SES) [45].

### 2.3. Statistical Analyses

We examined the dose–response relationship between the maternal prenatal depressive scores and the offspring’s internalizing, externalizing, or total difficulties problems by a smoothing plot. The logistic regression model was used to assess the independent association between the maternal prenatal depressive symptoms and the odds of the toddler’s internalizing, externalizing, or total difficulties problem accounted for the potential cofounders. We tested moderation by maternal pre-pregnancy BMI and GWG by introducing into the regression models the interaction terms of maternal depressive symptoms × pre-pregnancy BMI and maternal depressive symptoms × GWG. Effect modification by pre-pregnancy BMI and gestational weight gain was tested using Wald *p*-values (α = 0.10) and examining the stratum-specific results. Furthermore, we analyzed the effects of maternal depressive symptoms on the child EBPs in different groups categorized by maternal pre-pregnancy BMI and GWG. Missing covariate data were input using multiple imputation (the number of imputed datasets was 20). Significance tests were two-tailed, and the significance level was set at 0.05. All statistical analyses were performed using SAS 9.4 software.

## 3. Results

### 3.1. Sample Characteristics

Of the 1216 participants, most women were aged 25~35 years (77.06%), 32.40% reported alcohol use before pregnancy, 16.61% women reported passive smoking, and 14.64% women manifested pregnancy complication. Among the 1216 mother–child pairs, approximately half of the mothers were primipara (55.92%) and child born to cesarean (56.17%), and 52.14% child were male. Among the 1216 women, 38.65% of pregnant women had sub-threshold depressive symptoms (scores of CES-D above 8 and less than 16), and 12.01% had depressive symptoms (scores of CES-D above 16). A total of 16.37% and 23.03% participants were underweight and overweight/obese before pregnancy, while 26.48% and 31.50% of women manifested inadequate GWG and excessive GWG, respectively (Table 1). Compared to 1494 mother–child pairs without being followed-up at postpartum 24 months, the 1216 mothers included in this study tended to be overweight/obese (23.03% vs. 15.33%), gained inadequate GWG (26.48% vs. 23.29%), and chose mixed feeding practice at postpartum 6 months (33.47% vs. 28.25%), while there was no significant difference in the other characteristics between groups (Appendix A; *p* < 0.01).

### 3.2. Association between Maternal Prenatal Depressive Symptoms and Toddler’s EBPs

There was a strong, positive dose–response association between the maternal prenatal depressive symptoms and the likelihood of the toddler’s EBPs (Figure 1; *p* < 0.01). After confounder adjustment, each 1-score increase in prenatal depressive symptoms was associated with a 1.03-fold (95% CI: 1.01–1.06), 1.05-fold (95% CI: 1.02–1.07) and 1.03-fold (95% CI: 1.01–1.05) increase in the likelihood of internalizing, externalizing and total difficulties problems in children, respectively. Compared to children born to mothers without depressive symptoms, children exposed to prenatal sub-threshold depressive symptoms were associated with 48% and 42% increased risks for child internalizing (95% CI: 1.06–2.07), externalizing (95% CI: 1.02–1.99), respectively, while child exposed to prenatal depressive symptoms were associated with 69%, 106%, and 59% increased risks for child internalizing (95% CI: 1.05–2.72), externalizing (95% CI: 1.30–3.25), and total difficulties (95% CI: 1.04–2.45) problems, respectively. Compared to children born to mothers with appropriate GWG, children exposed to inadequate GWG were associated with 47% increased risks for child externalizing problems (95% CI: 1.01–2.13) (Table 2).

### 3.3. Modified Effects of Maternal Pre-Pregnancy BMI and GWG in the Association of Prenatal Depressive Symptoms and Child EBPs

Maternal pre-pregnancy BMI and GWG significantly modified the effects of maternal prenatal depressive symptoms on the likelihood of child externalizing and total difficulties problems (*p* < 0.10 for interaction), but there were no significant modified effects in the associations between maternal prenatal depressive symptoms and child internalizing problems (*p* > 0.10 for interaction) (Table 2). Further stratified analysis showed that among mothers with normal BMI before pregnancy, maternal prenatal depressive symptoms were associated with 1.93 times (95% CI 1.08–3.46) the risks of child externalizing problems compared to those in the non-depressive symptoms group (*p* = 0.037). However, among mothers with overweight/obese BMI before pregnancy, the maternal prenatal depressive symptoms were associated with 2.75 times (95% CI 1.06–7.11) the risks of child externalizing problems compared to those in the non-depressive symptoms group (*p* = 0.037) (Table 3). Among the women who gained inadequate GWG, maternal prenatal sub-threshold depressive symptoms were associated with higher risks for child externalizing problems (OR 2.85, 95% CI 1.48–5.48), and depressive symptoms were associated with higher risks for child externalizing and total difficulties problems (OR 4.87, 95% CI 2.03–11.70 and OR 2.94, 95% CI 1.28–6.74, respectively) than those in the non-depressive symptoms group (Table 3).

## 4. Discussion

In the prospective cohort study, we investigated the effects of prenatal depressive symptoms on the toddler’s EBPs at 24 months of age, and the modified effects of pre-pregnancy BMI and GWG on the association above. Our data showed stronger effects of maternal prenatal depressive symptoms on the risk for toddler’s internalizing, externalizing, and total difficulties problems. The main novel findings of the present study are that maternal sub-threshold depressive symptoms were significantly associated with higher internalizing and externalizing problems in children, and children were more likely to have higher risks for externalizing problems among women with overweight/obese before pregnancy and women who gained inadequate GWG during pregnancy.

Existing studies have mainly focused on the association of maternal prenatal depressive symptoms with the neurodevelopmental problems of school-age children and adolescents [11,12], without considering the sub-threshold depressive symptoms of pregnant women and offspring problem behavior in early childhood. It was estimated from previous study that the prevalence of subthreshold depressive symptoms ranged from 1.4% to 17.2% in adults in community settings [19], which indicated that numerous pregnant women tended to exhibit emotional symptoms during pregnancy instead of severe psychiatric disorders. Existing research has also found that the overall impact on the mortality of sub-threshold depressive symptoms and major depression disorders were comparable, with the population attributable fraction being 7.0% [19]. In the current study, we found that both maternal prenatal sub-threshold depressive symptoms and depressive symptoms were associated with increased risk of internalizing and externalizing problems in children aged 24-months, which was similar to a previous prospective cohort study that showed that children of mothers experiencing subclinical depressive symptoms (the scores of the Edinburgh Postnatal Depressive symptoms Scale range from 6 to 8) were at least two times more likely to have emotional-behavioral difficulties than the children of mothers reporting minimal symptoms [20]. Overall, women with sub-threshold depressive symptoms and not merely major depression during pregnancy should not be ignored considering the detrimental effects on the early life neurodevelopmental problems of offspring. 

Maternal abnormal BMI and depressive symptoms are major complications during pregnancy and are associated with severe health risks for their offspring including increasing the risk of preterm birth, suboptimal physical, cognitive and socio-emotional development, poorer academic performance, and physical and mental disorders in later life [25,46,47]. Previous studies have shown that maternal pre-pregnancy BMI, inappropriate GWG, and depressive symptoms are associated with adverse child neurodevelopment independently [48], however, the interaction effects of maternal overweight/obese, GWG, and depressive symptoms have not been tested yet. Recently, a comprehensive review demonstrated that maternal obesity and depression could increase the morbidity of child mental health problems, and speculated that their comorbidity may cause more adverse health outcomes for both the mothers and children [25]. Consistent with previous assumption, a novel finding of the current study was that the maternal prenatal depressive symptoms were associated with higher risk for child externalizing problems among women with overweight/obese BMI before pregnancy or gained inadequate GWG during pregnancy. The possible explanations may be related to the vulnerability of depressive symptoms in pregnant women that are overweight/obese [49]. A large epidemiological study also found that most of the somatic diseases such as metabolic diseases were associated with both self-reported and physician-diagnosed depressive symptoms, even after taking into account the role of gender, age and study, anxiety disorders, and somatic comorbidity [50]. The existing fact is that maternal prenatal depressive symptoms and abnormal BMI/GWG influence the children’s neurodevelopment via similar biological mechanisms such as low-grade inflammation [27,51] hypothalamic–pituitary–adrenal (HPA) axis deregulation [52], and increasing insulin resistance [53,54]. Additionally, the maternal prenatal weight condition and depressive symptoms could influence fetoplacental dysfunction, which in turn affect the secretory and transport function of placental and the later growth and development of the fetus [55,56]. Taken together, our findings fill in the gap that among different weight status before and during pregnancy, maternal depressive symptoms impact offspring neurodevelopment.

Our study has some strengths. First, to the best of our knowledge, this is the first study to explore the interaction effect of maternal weight status and depressive symptoms during pregnancy on toddler’s EBPs. Second, the current study found detrimental effects of maternal prenatal sub-threshold depressive symptoms (8 to 15 points of the CES-D scale) on child EBPs, which provide insights into efforts to reduce prenatal maternal depressive symptoms even without meeting the threshold of depressive symptoms. Third, based on the prospective cohort, reporting bias may be less common, and we adjusted for potential confounders influencing child neurobehavior including sociodemographic characteristics, maternal sleep problem, alcohol and smoke exposure, and feeding behaviors, thus, the results of the data are reliable. Fourth, the present study extends the evidence that both maternal abnormal pre-pregnancy BMI and GWG have interaction effects with prenatal depressive symptoms on the EBPs in children in early life, which extends and provides scientific basis for early identification and intervention considering the two factors. There were also some limitations. First, the mother’s pre-pregnancy BMI was calculated using the height and weight reported by pregnant women in the first trimester, and there may be a certain recall bias, however, a high correlation of self-reported and measured pre-pregnancy weights has been reported [57]. Second, maternal depressive symptoms were evaluated via the screening scale rather than clinical diagnosis, so further research is needed to replicate our results by evaluating the maternal depressive symptoms during pregnancy through various methods such as the use of prior medical archives, clinical interviews, structured or semi-structured diagnostic interviews, test scores, and simultaneous administration [58]. Third, mothers with depressive symptoms were more likely to report neurodevelopmental problems in toddlers, which may cause a certain degree of bias. However, we adjusted for multiple covariates and confounders that potentially influenced the child behavior problems. Future studies are needed to verify the combined effect of maternal abnormal weight condition and depressive symptoms on various offspring neurodevelopmental problems in a larger population and to explore the biological mechanisms and pathways involved.

## 5. Conclusions

Both maternal prenatal depressive symptoms and sub-threshold depressive symptoms were associated with increased risk of child internalizing and externalizing problems at 24 months old, and interacted with maternal pre-pregnancy BMI and GWG. Particularly, among the women who were overweight/obese before pregnancy or gained inadequate GWG, maternal prenatal depressive symptoms had higher risks of child externalizing problems. Our study highlights the importance of simultaneously controlling the weight of pregnant women before pregnancy and prompting mental health in pregnant women, which might be of benefit to the offspring EBPs.

## Figures and Tables

**Figure 1 nutrients-15-00181-f001:**
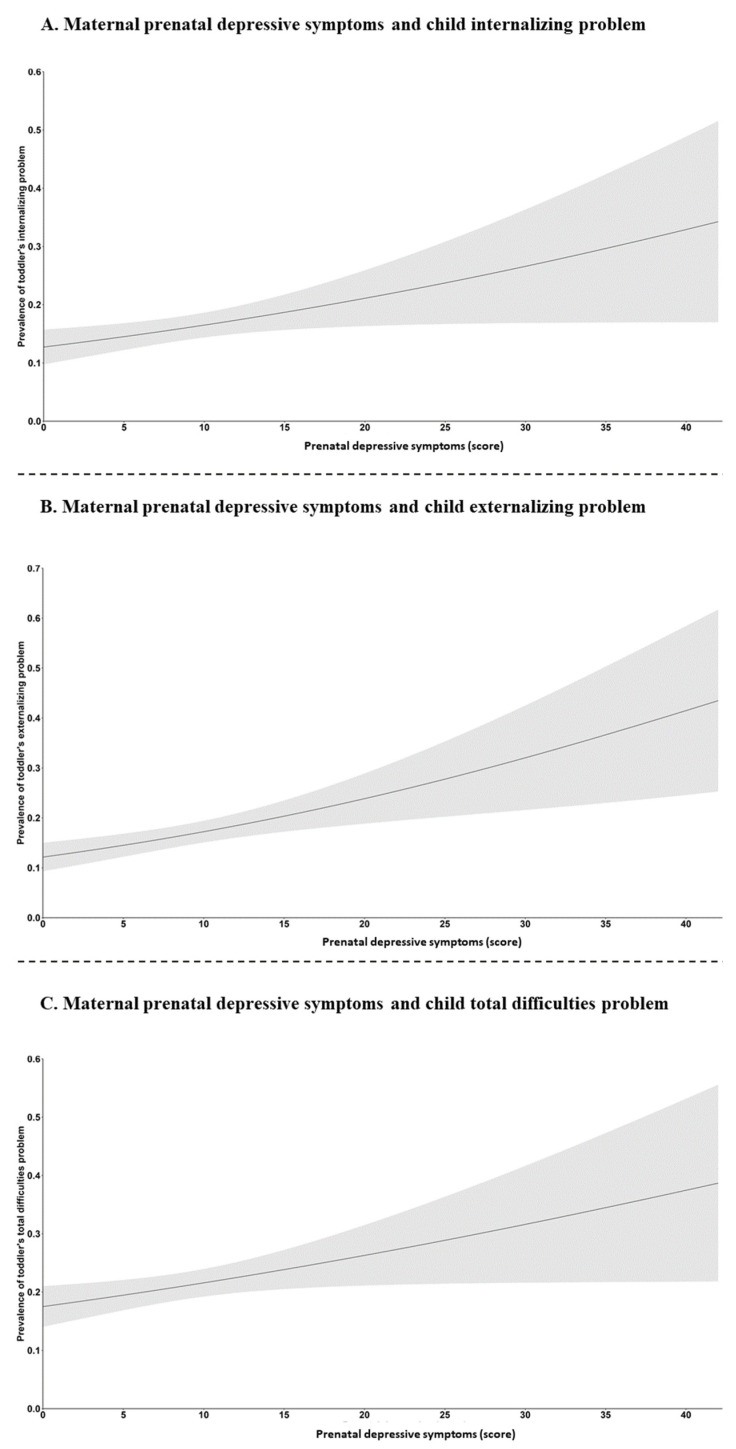
Crude association between the maternal prenatal depressive symptoms and the likelihood of toddler’s EBPs at 2 years of age. Note: The solid line represents the point estimate and the grey area represents the 95% confidence bands around the estimate (all *p* < 0.01). Curves were estimated by calculating the predicted probabilities based on an unadjusted logistic regression model. (**A**–**C**) The unadjusted association between the maternal prenatal depressive symptoms and the likelihood of toddler’s internalizing problem, externalizing problem, and total difficulties problem at 2 years of age.

**Table 1 nutrients-15-00181-t001:** Demographic characteristics in the mother–child pairs.

Characteristics	*n* (%)/Mean ± SD
Maternal variables	
Age groups, years	
<25	156 (12.83)
25~35	937 (77.06)
≥35	123 (10.12)
Social economic status	
High	550 (45.23)
Low	666 (54.77)
Pre-pregnancy alcohol use	
Yes	394 (32.40)
No	822 (67.60)
Passive smoking during pregnancy	
Yes	202 (16.61)
No	1014 (83.39)
Pregnancy complications	
Yes	178 (14.64)
No	1038 (85.36)
Parity	
Primipara	680 (55.92)
Multipara	536 (44.08)
Delivery mode	
Cesarean	683 (56.17)
Vaginal	533 (43.83)
Pre-pregnancy BMI, kg/m^2^	21.54 ± 3.14
<18.5	199 (16.37)
18.5~24	737 (60.61)
≥24.0	280 (23.03)
GWG, kg	13.72 ± 5.48
Inadequate GWG	322 (26.48)
Appropriate GWG	511 (42.02)
Excessive GWG	383 (31.50)
Prenatal depressive scores, score	8.70 ± 6.15
Non-depressive symptoms	600 (49.34)
Sub-threshold depressive symptoms	470 (38.65)
Depressive symptoms	146 (12.01)
Offspring’s variables	
Sex	
Male	634 (52.14)
Female	582 (47.86)
Preterm	
Yes	38 (3.13)
No	1178 (96.88)
Breastfeeding practice at 6 months old	
Exclusive breastfeeding	525 (43.17)
Mixed feeding	407 (33.47)
Formula feeding	284 (23.36)
Internalizing problem at 2 years old	6.13 ± 2.40
Yes	196 (16.12)
No	1020 (83.88)
Externalizing problem at 2 years old	6.28 ± 2.49
Yes	204 (16.78)
No	1012 (83.22)
Total difficulties at 2 years old	12.42 ± 4.18
Yes	257 (21.13)
No	958 (78.87)

Note: SD, standard deviation; BMI, body mass index; GWG, gestational weight gain.

**Table 2 nutrients-15-00181-t002:** Association between the maternal prenatal depressive symptoms and the likelihood of toddler’s EBPs. Odds ratios are shown for a 1-score increase in the prenatal depressive symptoms scores as well as for representative prenatal depressive symptoms compared with the non-depressive symptoms.

Variables	Internalizing Problem			Externalizing Problem			Total Difficulties		
	*n* (%)	Model 1	Model 2	*n* (%)	Model 1	Model 2	*n* (%)	Model 1	Model 2
Prenatal depressive symptoms, 1-score increase	—	1.03 (1.01, 1.06)	1.03 (1.01, 1.06)	—	1.04 (1.02, 1.06)	1.05 (1.02, 1.07)	—	1.03 (1.01, 1.05)	1.03 (1.01, 1.05)
Prenatal depressive symptoms									
Non-depressive symptoms	81 (13.50)	Ref.	Ref.	83 (13.83)	Ref.	Ref.	111 (18.50)	Ref.	Ref.
Sub-threshold depressive symptoms	86 (18.30)	1.43 (1.03, 1.99)	1.48 (1.06, 2.07)	86 (18.30)	1.39 (1.01, 1.94)	1.42 (1.02, 1.99)	108 (22.98)	1.31 (0.98, 1.77)	1.33 (0.99, 1.80)
Depressive symptoms	29 (19.86)	1.64 (1.02, 2.62)	1.69 (1.05, 2.72)	35 (23.97)	1.99 (1.28, 3.13)	2.06 (1.30, 3.25)	38 (26.03)	1.58 (1.03, 2.42)	1.59 (1.04, 2.45)
Pre-pregnancy BMI									
Underweight	39 (19.60)	1.22 (0.80, 1.87)	1.20 (0.78, 1.85)	34 (17.09)	0.94 (0.61, 1.46)	0.97 (0.61, 1.52)	47 (23.62)	1.09 (0.74, 1.61)	1.08 (0.72, 1.61)
Normal weight	122 (16.55)	Ref.	Ref.	131 (17.77)	Ref.	Ref.	160 (21.71)	Ref.	Ref.
Obese	35 (12.50)	0.72 (0.48, 1.09)	0.73 (0.48, 1.11)	39 (13.93)	0.75 (0.50, 1.10)	0.76 (0.51, 1.13)	50 (17.86)	0.79 (0.55, 1.13)	0.81 (0.56, 1.16)
GWG									
Inadequate GWG	57 (17.70)	1.08 (0.74, 1.58)	1.11 (0.75, 1.63)	65 (20.19)	1.47 (1.01, 2.13)	1.40 (0.95, 2.03)	78 (24.22)	1.26 (0.90, 1.77)	1.24 (0.87, 1.75)
Appropriate GWG	82 (16.05)	Ref.	Ref.	75 (14.68)	Ref.	Ref.	102 (19.96)	Ref.	Ref.
Excessive GWG	57 (14.88)	1.00 (0.68, 1.46)	1.02 (0.69, 1.51)	64 (16.71)	1.21 (0.83, 1.76)	1.19 (0.81, 1.75)	77 (29.96)	1.07 (0.76, 1.50)	1.04 (0.74, 1.49)
Maternal depressive symptoms × pre-pregnancy BMI, *p*-value		0.392	0.305		0.004	<0.001		0.088	0.040
Maternal depressive symptoms × GWG, *p*-value		0.947	0.836		0.040	0.020		0.137	0.090

Note: OR, odds ratio; CI, confidence interval. Model 1: without adjusted any confounders. Model 2: adjusted for maternal age, social economic status, pre-pregnancy alcohol use, passive cigarette smoking, pregnancy complications, parity, delivery mode, offspring’s sex, preterm, breastfeeding practice at 6 months old.

**Table 3 nutrients-15-00181-t003:** Modified effects of pre-pregnancy BMI and GWG on the association between maternal prenatal depressive symptoms and toddler’s externalizing and total difficulties problems.

Variables	Externalizing Problem			Total Difficulties		
	*n* (%)	aOR (95% CI)	*p*	*n* (%)	aOR (95% CI)	*p*
Pre-pregnancy BMI groups						
Underweight						
Prenatal depressive symptoms, 1-score increase	—	1.05 (0.99, 1.11)	0.127		1.01 (0.96, 1.07)	0.663
Prenatal depressive symptoms						
Non-depressive symptoms	13 (12.62)	Ref.		22 (21.36)	Ref.	
Sub-threshold depressive symptoms	18 (22.50)	2.21 (0.96, 5.08)	0.063	21 (26.25)	1.23 (0.60, 2.52)	0.574
Depressive symptoms	3 (18.75)	2.13 (0.48, 9.49)	0.319	4 (25.00)	1.30 (0.35, 4.84)	0.697
Normal weight						
Prenatal depressive symptoms, 1-score increase	—	1.04 (1.01, 1.07)	0.016		1.03 (1.01, 1.06)	0.023
Prenatal depressive symptoms						
Non-depressive symptoms	57 (14.92)	Ref.		66 (18.44)	Ref.	
Sub-threshold depressive symptoms	57 (18.63)	1.32 (0.87, 2.00)	0.188	70 (24.14)	1.42 (0.97, 2.10)	0.075
Depressive symptoms	22 (23.16)	1.93 (1.08, 3.46)	0.026	24 (25.97)	1.68 (0.97, 2.92)	0.064
Overweight/obesity						
Prenatal depressive symptoms, 1-score increase	—	1.12 (1.05, 1.19)	<0.001		1.01 (0.96, 1.07)	0.663
Prenatal depressive symptoms						
Non-depressive symptoms	16 (11.51)	Ref.		23 (16.55)	Ref.	
Sub-threshold depressive symptoms	11 (13.00)	1.01 (0.44, 2.36)	0.975	17 (17.86)	0.94 (0.45, 1.95)	0.849
Depressive symptoms	10 (25.64)	2.75 (1.06, 7.11)	0.037	10 (28.57)	1.77 (0.73, 4.30)	0.071
GWG groups						
Inadequate GWG						
Prenatal depressive symptoms, 1-score increase	—	1.09 (1.04, 1.14)	<0.001		1.06 (1.01, 1.10)	0.011
Prenatal depressive symptoms						
Non-depressive symptoms	19 (11.73)	Ref.		31 (19.14)	Ref.	
Sub-threshold depressive symptoms	32 (26.89)	2.85 (1.48, 5.48)	0.002	33 (27.73)	1.60 (0.89, 2.89)	0.116
Depressive symptoms	14 (34.15)	4.87 (2.03, 11.70)	<0.001	14 (34.15)	2.94 (1.28, 6.74)	0.011
Appropriate GWG						
Prenatal depressive symptoms, 1-score increase	—	1.03 (0.99, 1.07)	0.166		1.02 (0.99, 1.06)	0.200
Prenatal depressive symptoms						
Non-depressive symptoms	36 (14.40)	Ref.		48 (19.20)	Ref.	
Sub-threshold depressive symptoms	28 (14.29)	1.07 (0.61, 1.86)	0.824	38 (19.39)	1.02 (0.63, 1.67)	0.926
Depressive symptoms	11 (16.92)	1.27 (0.55, 2.59)	0.653	16 (24.62)	1.29 (0.66, 2.52)	0.453
Excessive GWG						
Prenatal depressive symptoms, 1-score increase	—	1.04 (0.99, 1.09)	0.095		1.01 (0.97, 1.06)	0.660
Prenatal depressive symptoms						
Non-depressive symptoms	28 (14.89)	Ref.		32 (17.02)	Ref.	
Sub-threshold depressive symptoms	26 (16.77)	1.15 (0.62, 2.15)	0.658	37 (23.87)	1.58 (0.90, 2.77)	0.113
Depressive symptoms	10 (25.00)	1.97 (0.82, 4.70)	0.127	8 (20.00)	1.17 (0.47, 2.87)	0.739

Note: BMI, body mass index; GWG, gestational weight gain; OR, odds ratio; CI, confidence interval. All models were adjusted for maternal age, social economic status, pre-pregnancy alcohol use, passive cigarette smoking, pregnancy complications, parity, delivery mode, offspring’s sex, preterm, and breastfeeding practice at 6 months old.

## Data Availability

Supporting data are not available online because the participants in this study did not agree for their data to be shared publicly.

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
