# Peer review of "Maternal Pre-Pregnancy BMI and Gestational Weight Gain Modified the Association between Prenatal Depressive Symptoms and Toddler’s Emotional and Behavioral Problems: A Prospective Cohort Study"

_nutrients, 2022, doi:10.3390/nu15010181_

Round 1

Reviewer 1 Report

This is an article presenting a topic of interest for the journal using a large sample.

Some points need to be clarified:

How was depression evaluated? Where there any additional psychiatric interviews or just the administration of a single questionnaire??  

In the text ‘depressive symptomatology’ should be used as a term instead of the term depression which needs clinical examination and not just the use of a questionniare. This point has been already raised, please add the criticism regarding depressive symptom assessment with tools such CES-D used in the current study as discussed in general in:  Giannouli, V. (2017). Fatigue, depression, and obesity in patients with Rheumatoid Arthritis; More questions than answers: Comment on the Article by Katz et al. Arthritis Care & Research69(3), 454-455.

In addition to that, a more detailed description of the recruitment should be made in the methods section.

The tables should be explained in detail in relevant text in the discussion. Overall, a very interesting study.

Reviewer 2 Report

I have read this paper with great interest, and highly value the effort. However, my main concern relates to the causality suggested. 

This is to be best of my understanding an association analysis, while some wording suggest causality. I highly recommend to adapt title, abstract (eg last sentence) and text on this aspect.

Methods

It is not clear yet to me how toddler recruitment has been done, while prenatal depression score were collected ? is this standard, or were toddler recruited from a previously existing cohort of pregnant women ?

Who has assessed the toddlers on EBP ? mothers ? trained researchers or others ?

Is the cohort representative for the population on its pregnancy characteristics ?

Artificial feeding = better us ‘formula’
